# Semi-Automatic Method for Early Detection of *Xylella fastidiosa* in Olive Trees Using UAV Multispectral Imagery and Geostatistical-Discriminant Analysis

Annamaria Castrignanò [1], Antonella Belmonte [2,*], Ilaria Antelmi [3], Ruggiero Quarto [4], Francesco Quarto [5], Sameh Shaddad [6], Valentina Sion [3], Maria Rita Muolo [7], Nicola A. Ranieri [7], Giovanni Gadaleta [8], Edoardo Bartoccetti [9], Carmela Riefolo [1], Sergio Ruggieri [1] and Franco Nigro [3]

[1] CREA-AA—Council for Agricultural Research and Economics, Via Celso Ulpiani, 5, 70125 Bari, Italy; annamaria.castrignano@unich.it (A.C.); carmela.riefolo@crea.gov.it (C.R.); sergio.ruggieri@crea.gov.it (S.R.)

[2] CNR-IREA National Research Council—Institute for Electromagnetic Sensing of the Environment, Via Amendola, 122/D, 70126 Bari, Italy

[3] Department of Soil, Plant and Food Sciences, University of Bari—Aldo Moro, Via G. Amendola 165/A, 70126 Bari, Italy; ilaria.antelmi@uniba.it (I.A.); valentina.sion@uniba.it (V.S.); franco.nigro@uniba.it (F.N.)

[4] Department of Earth and Geo-Environmental Sciences, University of Bari—Aldo Moro, Via Edoardo Orabona, 4, 70125 Bari, Italy; ruggiero.quarto@senato.it

[5] PRO-GEO s.a.s., Via M. R. Imbriani 13, 76121 Barletta, Italy; progeosas@alice.it

[6] Soil Science Department, Faculty of Agriculture, Zagazig University, 44511 Zagazig, Egypt; SMShaddad@agri.zu.edu.eg

[7] Servizi di Informazione Territoriale S.r.l., Piazza Giovanni Paolo II, 8, 70015 Noci, Italy; mr.muolo@sit-puglia.it (M.R.M.); nicola.ranieri@sit-puglia.it (N.A.R.)

[8] Professional Agronomist, Via Carr. Lamaveta, 63/F, 76011 Bisceglie, Italy; gadaleta73@epap.sicurezzapostale.it

[9] Salt&Lemon, Piazza Mascagni 11, 10015 Ivrea, Italy; edoardo.bartoccetti@saltlemon.it

[*] Correspondence: belmonte.a@irea.cnr.it

**Abstract:** *Xylella fastidiosa* subsp. *pauca* (*Xfp*) is one of the most dangerous plant pathogens in the world. Identified in 2013 in olive trees in south–eastern Italy, it is spreading to the Mediterranean countries. The bacterium is transmitted by insects that feed on sap, and causes rapid wilting in olive trees. The paper explores the use of Unmanned Aerial Vehicle (UAV) in combination with a multispectral radiometer for early detection of infection. The study was carried out in three olive groves in the Apulia region (Italy) and involved four drone flights from 2017 to 2019. To classify *Xfp* severity level in olive trees at an early stage, a combined method of geostatistics and discriminant analysis was implemented. The results of cross-validation for the non-parametric classification method were of overall accuracy = 0.69, mean error rate = 0.31, and for the early detection class of accuracy 0.77 and misclassification probability 0.23. The results are promising and encourage the application of UAV technology for the early detection of *Xfp* infection.

**Keywords:** change of support; polygon (co)kriging; canonical analysis; uncertainty

## 1. Introduction

The bacterium *Xylella fastidiosa* is considered one of the most dangerous plant pathogens in the world, and has already caused diseases in more than 300 species in Brazil and U.S., with serious economic losses. In Europe, *X. fastidiosa* subsp. *pauca* (*Xfp*) was identified in 2013 in the Apulia region (south–eastern Italy), renowned in the world for its olive oil production, and have already caused very serious losses to oliviculture, one of the main productive sectors of the region, with a remarkable impact on the environment, the landscape, and the cultural heritage of that territory [1]. *Xfp* is a Gram-negative bacterium that lives and reproduces into the xylem, thus clogging the conductive vessels of the sapwood carrying water and mineral nutrients. The symptoms caused by *Xfp* infection in the olive

trees are the intense browning of the leaves (leaf scorch) and more or less extensive drying of the foliage flap. The phenomenon affects, first, ramifications, and then small branches of the foliage, isolated and distributed at random, especially starting from the upper stands, and then spreading to the entire branches, until it reaches an extension that affects the whole aerial portion of the plant. Other symptoms are the reduced growth of branches and shoots. In the final stage, the foliage assumes a burnt color, but the tree remains alive, although is emaciated in its external aspect, and its roots, as long as they remain vital, are still able to emit suckers destined to dry in a short time.

The pathogen is currently spreading, and a serious risk exists that the bacterium may affect the whole of Europe, particularly all of the countries of the Mediterranean Basin [2]. The attention of producers and consumers to the largest area of olive cultivation in the world is therefore understandable: more than 2.5 million hectares in the Mediterranean basin. Unfortunately, at present, there is no cure, and the only solution to stop the epidemic spread of the disease is to pluck infected trees. A disease management strategy would be much more efficient if the asymptomatic or infected plants with visible symptoms of desiccation were identified at an early stage, to reduce the spreading of the pathogen and the risk of infection to neighboring trees [3–5]. Disease severity on olive plants can be quantified in several ways (occurrence, intensity, severity level) and at different scales (leaves, stems, fruits, portion/whole plants, or small quadrats). At present the most common approach is visual rating, which is now reasonably well understood and its practice is over 100 years old [6,7]. Undoubtedly, it has some advantages: the process can be quick; it may be relatively easy to recognize the disease or differentiate multiple diseases with proper training, and no expensive equipment is required, although the use of technological aids can improve the results [8]. In the framework of the mandatory control program of quarantine pathogens managed by the regional Phytosanitary Authority, the monitoring of *Xfp* is currently based on the "visual inspection" of the plants, i.e., searching for visible symptoms on the canopy of the trees. However, visually assessed disease may be affected by variability in accuracy (subjectivity) and need to develop standards to aid assessment and/or to repeat training, maintaining quality. Therefore, visual rating may be time-consuming, expensive, and destructive if samples are collected in the field to be later analyzed in the laboratory [8].

It would then be desirable to have an automatic method for the detection of the pathogen that is fast, reliable, relatively inexpensive and allows real-time monitoring of the disease for the control and precise management of the infection. Remote sensing technologies have been used for a long period to identify and monitor disease [5,9–12]. However, the poor revisiting times and/or coarse spatial resolutions limit their use in the disease detection and the symptoms assessment, in which it is necessary to intervene promptly in the early stages of the infection to mitigate the damage caused by the spread of the pathogen. Unmanned aerial vehicles (UAVs), also known as drones, have already been used in some agricultural procedures; however, in the last few years they have expanded very rapidly, especially in the field of precision farming [13–15]. UAVs are capable of covering large areas to be monitored much faster than people on the ground can do; therefore, they can be used as a very efficient tool for scouting.

Recent work on *Xfp* in olive trees [5,12,16] has demonstrated that infection causes a change in spectral reflectance. Actually chlorophyll content tends to decrease in infected plants, which causes a higher reflectance in the visible region (VIS) and a blue shift in the red-edge portion of the spectrum (700–720 nm) [3,17–21]. Moreover, stressed plants show a reduction in canopy density and leaf area leading to a decrease of spectral reflectance in the near-infrared region (NIR, 680–800 nm) [3]. It then makes possible using UAVs, in combination with cost-efficient and light-weight multispectral sensors operating in the VIS, red-edge, and NIR regions, for disease infection before it becomes widespread. However, to transform UAV into an effective tool for disease management, it needs to extract useful information from multispectral images. Currently, many image processing,

machine learning, and linear discriminant analysis (LDA) techniques have already been sufficiently developed to extract information useful for specific purposes [3,22–25].

However, when dealing with spatial data in a multivariate approach, it is crucial to face a major problem, which is unfortunately still much little considered in scientific literature, which is the one of their support. The support of a spatial variable is the physical volume over which the value of the variable is measured or computed [26]. In remote sensing, the term "pixel" is practically interchangeable with it. Changing the support of a variable through averaging or assimilation creates a new variable, which is related to the original one, but with different statistical and spatial characteristics [27]. The problem of determining how these properties vary with the support is called the change of support problem (COSP) [28] and this must be taken into account when jointly analyzing data with different support in a multivariate analysis. Geostatistics has proposed possible solutions to this problem [29,30], mostly based on block (co)kriging or on its more flexible version that is better suited to the Geographic Information System (GIS) context, known as polygon (co)kriging [27].

The real novelty of this article is to propose and apply an integrated approach of various statistical and spatial geostatistical techniques taking into account change of support, and able to discriminate asymptomatic and/or infected plants, but at a very early stage, by using radiometric data collected with a multispectral sensor on board of a drone.

In particular, the study describes a semi-automatic combined method of geostatistics and discriminant analysis, to classify the *Xfp* symptom severity level on olive trees with a special focus on early detection.

## 2. Materials and Methods

### 2.1. Study Site Description

The agronomic surveys involved two study areas located at Oria and Torchiarolo (Brindisi province, south–eastern Italy, Figure 1), and three olive orchards (40°31′12″ N, 17°39′36″ E; 40°31′16″ N, 17°39′40″ E; 40°30′36″ N, 18°03′36″ E), where the evolution and spread of the quick *Xfp* syndrome of olive trees (*Olea europaea* L.) were monitored.

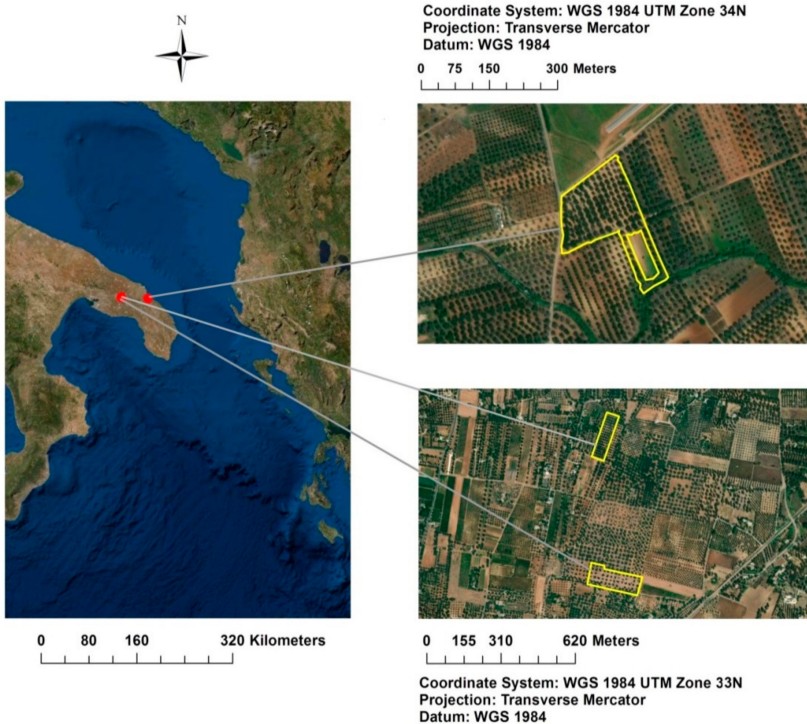

**Figure 1.** Localization of the three olive groves in the Apulia region (south–eastern Italy): Re field, Fella field, Torchiarolo field.

A short description of the study sites follows:

Oria (Re field): the olive grove consists of a single portion of centenarian trees of the "Ogliarola Salentina" cultivar. The trunk is large, twisted, with a dense and shrubby foliage. The planting system is very wide and in some respects irregular; there is no irrigation system. Periodic ploughing and mulching operations were carried out in the plot in order to contain weeds.

Oria (Fella field): the olive grove consists of a single portion of 50–60 years old trees of the "Cellina di Nardò" cultivar. The trunk is regular, fairly linear and columnar in shape, with a thick and shrubby canopy. The planting system is wide and regular and there is no irrigation system. Periodic ploughing was carried out on the plot in order to contain weeds.

Torchiarolo field: the olive grove is made up of a portion of centenarian trees of the "Cellina di Nardò" cultivar, and of a neighboring portion of olive trees of the same cultivar surrounding a trellis-shaped vineyard. The trunk is large in size, fairly linear in shape; the canopy is dense. The planting system is wide and regular and there is no irrigation system. Mechanical operations were carried out on the two plots for weeding. The field, which has a regular shape, is bounded on all four sides by a large uncultivated field, a high road, a waste-water drainage canal, and another olive grove with similar characteristics.

*2.2. Variables*

The following variables were used for the construction of the prediction model:

Response variable: visual assessment of disease severity level.

A survey was carried out on each olive plant in order to evaluate the level of *Xfp* disease severity. The visual inspection of the symptoms with the relative photographic documentation was carried out by an operator who scored the percentage of affected canopy, examining the portion of foliage with wilting symptoms in each of the four cardinal points.

The plants were grouped according to an empirical scale of symptom severity with increasing values from 1 (asymptomatic plants) to 6 (plants characterized by lack of green parts and, therefore, dead), as follows:

Class 1, asymptomatic: 0% foliage with drying symptoms.

Class 2, slight wilting severity: 1% to 20% foliage with drying symptoms.

Class 3, medium–low severity: 20% to 50% foliage with drying symptoms.

Class 4, medium–high severity: 50% to 70% foliage with drying symptoms.

Class 5, high severity: 70% to 99% foliage with drying symptoms.

Class 6, dead or without green vegetation: presence of symptoms >99% (Figure 2).

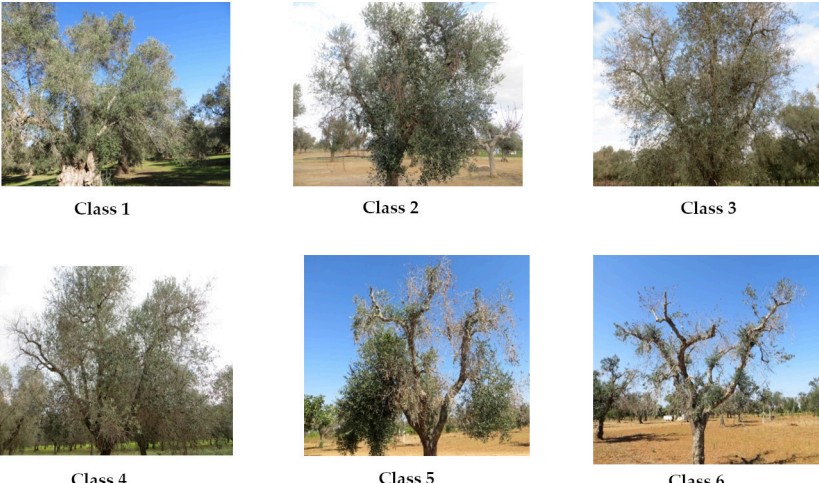

**Figure 2.** Examples of the six classes of *Xylella fastidiosa* subsp. *pauca* (*Xfp*) severity level.

Therefore, the evaluation of the wilting severity does not refer to the entire plant, but to an angular portion of the canopy of 90° aperture, according to the four cardinal directions, which is to be assumed as the support of this measure.

However, this work was aimed more specifically at identifying asymptomatic plants or in the very early stage of the disease, as any delay would preclude the effectiveness of the measures to slow down the epidemic progression and increase the infection risk for the surrounding plants. This aspect is particularly relevant for *Xfp*, whose latency period on olive trees (i.e., the time lapse from the infection to the symptoms appearance) can last 12–18 months. The "ideal" early detection stage would be located into the "latency period", when the symptoms are still not visible. However, this is not possible, being the official monitoring activity based on the "visual inspection" of the plant, i.e., searching for wilting symptoms on the canopy of the tree. Therefore, only two classes of wilting severity were considered: Class 0, including asymptomatic plants or plants with a wilting severity in each angular portion of the canopy not exceeding 5%, and Class 1, including all remaining plants.

The choice of this threshold value was based on the experience of well-trained people in visual rating.

Quantitative variables as predictors: UAV multispectral data.

The UAV used in this study was a multi-rotor DJI Mavic Pro drone. The maximum payload capacity was ~800 g, and the maximum flight range 7 km. The vehicle was equipped with a three-axis accelerometer, a gyroscope, an integrated barometric precision sensor for altitude control, and with four brushless motors powered by a battery. It flew autonomously with the aid of its Global Position System (GPS) receiver and its waypoint navigation system. The drone endurance with the designed payload was 12/13 min.

A custom payload tray was designed to carry the multispectral sensor (Parrot Sequoia) that consisted of a four-band multispectral camera at high resolution (16 megapixel), with the wavelengths centered in green (550 ± 20 nm), red (660 ± 20 nm), red-edge (735 ± 5 nm), and near infrared (790 ± 20 nm). The choice of these particular bands, instead of the usual RGB, is due to the well-known utility of the red-edge as a general indicator of plant stress, and the NIR as an indicator of the structure and leaf area of the canopy [3,17], which was one of the features of the olive tree most sensitive to infection.

The high spatial resolution UAV images were acquired on September 2017, March 2018, June 2018, and June 2019 for Re and Fella fields, whereas for Torchiarolo field, the last date was August 2019, at the height of 70 m, and with the theoretical ground sampling distance (GSD) of 6.6 cm/pixel; see Figure 3, as an example.

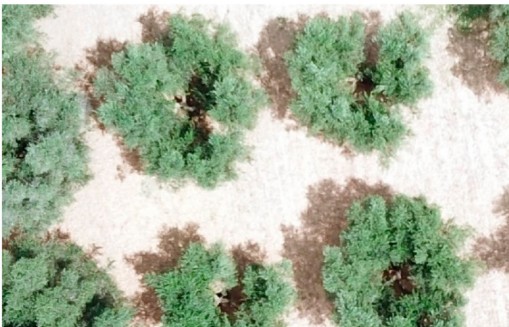

**Figure 3.** High resolution frame portion acquired by Unmanned Aerial Vehicle (UAV) at Oria field on September 2017.

As the datasets where collected and processed over a long period of time, different versions of Pix4d Mapper software were used. The software was always kept updated to the latest version.

However, after all flights were completed (August 2019), all of the spectral datasets were reprocessed with the version 4.4.12 of Pix4d Mapper. The software automatically manage the layers of the different bands, as the camera rig is already recognized in the software database. In order to have exactly the same areas (with the same extent and pixel

number) for each flight and for each band, some target on the ground, which well-known coordinates are used by Pix4d Mapper software. For each elaboration, the same shapefiles were used to crop elaboration areas, and upscaled output ground sampling distance to the ones with the highest values. The software is a proprietary closed software so no further information about its internal algorithms are provided.

The software uses Structure from Motion (SfM) techniques to reconstruct the scene, based on a large number of overlapping photos, to generate a final orthophoto image from a quoted point cloud.

### 2.3. Methodology

The construction of the prediction model consisted of the following main steps:

1. Extraction of canopies for each field from UAV images.

The data processing, for the semi-automatic extraction of olive trees crowns, is depicted in the following main tasks.

✓ Pre-processing:
- To create a composite multispectral image from the individual spectral bands, the procedure of layer stacking has been applied, so individual image bands to have the same extent (no. of rows and columns).
- To enhance specific information about the landscape, which cannot easily be seen with a natural color image, as stress and vigor of vegetation, the false color composite has been applied. In this case, to emphasize the status of the plant, the following bands have been combined: Red → Red, Red-Edge → Green, Green → Blue, (Figure 4a).

✓ Supervised Classification:
- Supervised classification was applied, which involves the use of training area data considered representative of the land cover types of the study areas. In this case three labels were identified: soil, canopy and shadow. As classifier algorithm, the maximum likelihood was used, which is based on maximizing the likelihood that the observed values follow a normal distribution (Figure 4b).

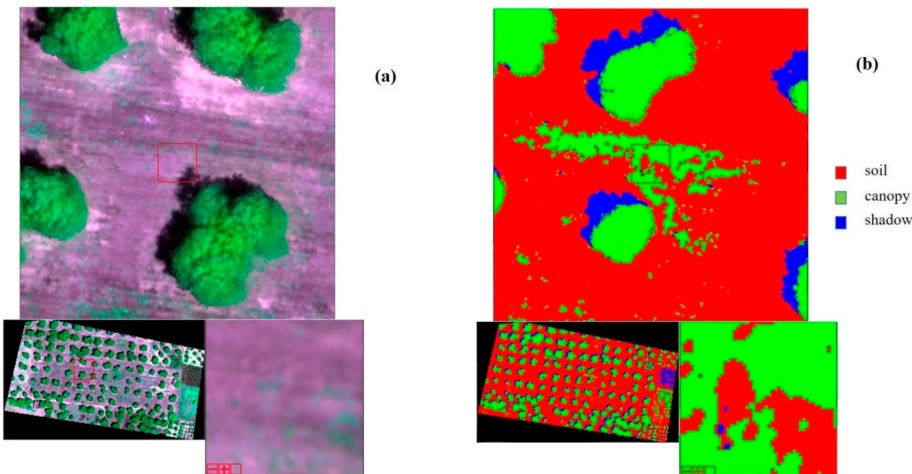

**Figure 4.** (**a**) An example, Oria Field: 4 layer stack at false colors. (**b**) An example, Oria Field: thematic map of classification.

- To smooth the boundaries of small areas located near each other, or to aggregate these areas, a morphological filter [31–33], which uses the fundamental operations of erosion and dilation, was preferred to fill gaps in the contour lines. Subsequently, a format conversion has been applied to convert labelled raster image into vector data to extract closed spatial features (polygons) from the classification.

✓　Exporting to GIS Environment:

- Each closed polygon, representing an individual plant, was then imported into GIS environment and an editing procedure was applied to generate a multi-polygon product (Figure 5). This process allowed to modify the vertices of the selected spatial feature, to fill eventual holes in the polygon and/or to cut some parts.
- Each polygon was further subdivided into four quadrants (North, East, South, and West), to which to refer both visual and radiometric measurements (common support). The quadrants were generated by a procedure that has been implemented in C# language with ArcObjects libraries to be integrated into the ArcGIS environment. This procedure is based on segments that join the centroid of each polygon with the four points at 45°, 135°, 225°, 315° (defined above the horizon), to split the polygon into the north, east, south, and west sectors of the crown, respectively.
- Each shapefiles of quadrants has been imported in geostatistical environment.

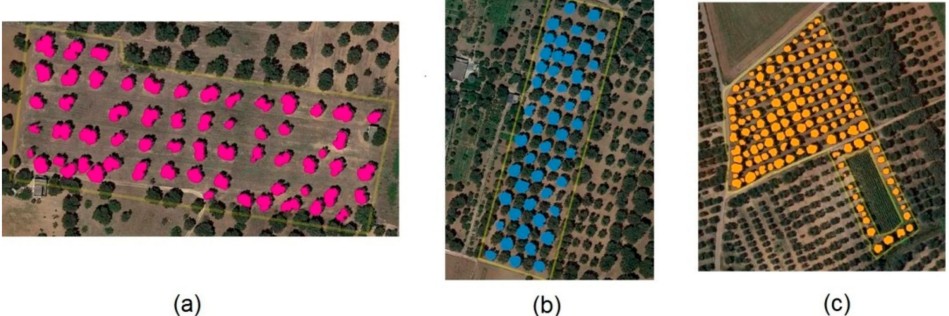

**Figure 5.** An example of multipolygon product of the three fields on the date of September 2017: Re (**a**), Fella (**b**), and Torchiarolo (**c**).

For the classification and extraction of the spatial features the software ENVI (Environment for Visualizing Images) 5.1 was used. Quantum GIS 3.8 was employed for extracting multipolygon product from the classification results and exporting to geostatistical environment.

Figure 6 summarizes the process of canopy extraction from UAV data.

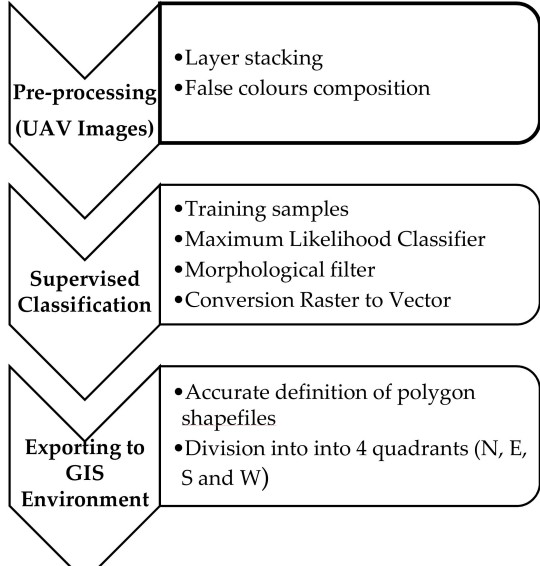

**Figure 6.** Main processing tasks of semi-automatic extraction of olive tree canopies.

2.  Change of support: polygon cokriging.

Since the UAV data had a very fine spatial resolution (about 0.07 m), the images were transformed to the support of single quadrant of each individual canopy, which was the support of the visual surveys. This change of support was necessary to jointly analyze these two types of data for the creation of the prediction model of infection severity level. At this end, polygon cokriging [27,34,35] was used that is an extension of block cokriging [36,37] when block (polygon) has an irregular and variable shape over space. In our case the variable block is represented by each quadrant of the previously extracted canopies of the olive trees. The expected values of UAV multi-band reflectivity and their standard deviations were then estimated over this support. Each polygon was first discretized in regular cells for the calculation of the areal (polygon) co-variance function, which was expressed as a weighted discrete summation of the point (pixel) covariance functions calculated at the centroid point of each cell. In this procedure, the weights correspond to the proportion of cell falling inside the polygon [38].

After that, the calculation proceeded according to the standard formulation of the block cokriging, as exhaustively described in several geostatistical manuals [36–38]. Here, it is only intended to underline that the spatial correlation structure of the multitemporal and multi-band UAV data for each field has been analyzed by adopting a linear model of co-regionalization (LMC) to the reflectivity data of the four bands recorded on the four, flight dates for a total of 16 variables.

LMC [39] considers all of the studied variables to be generated by the same independent physical processes acting at $N_s$ different spatial scales [40–42]. All (both direct and cross-) variogram models are expressed as linear combinations of the same basic structures for each spatial scale (range), represented by variograms standardized to unit sill, and with the coefficients equal to partial sills [41]. These last ones reflect the influence of the specific spatial scale on the total spatial variation of the study variable.

All geostatistical analyses were performed using the software ISATIS (Geovariances, France, 2017).

### 2.4. Construction of the Prediction Model: Statistical Analyses

Moreover, for the statistical analysis, finalized to the construction of the prediction model, various phases can be considered, including different procedures, to be applied in a consecutive and integrative rather than alternative way.

1.  Selection of predictors. The first step in the statistical analysis was preliminary to determine which variables (bands) were significantly related to the response class variable (disease severity level), coded as shown above. For this purpose, a stepwise discriminant analysis was performed, which is a regression technique aimed to select a subset of the quantitative variables (predictors) for use in discriminating between the classes. The variables are chosen to enter or to leave the regression model on the basis of the significance level of an F test from an analysis of covariance, where the variables already chosen act as covariates and the variable under consideration is the response variable [39]. The analysis was performed using the STEPDISC procedure of SAS software in stepwise mode [39], and the significance levels for a variable to enter the subset and to stay in the subset were set to 0.1.

2.  Check of multivariate normality. The next step for implementing discriminant analysis was to check the multivariate normality in each of the two severity classes of *Xfp* symptoms, since the estimation of misclassification probabilities requires the assumption of multivariate normality. Since the condition of normality for each of the 16 quantitative variables is a necessary, but not sufficient, condition for multinormality, first, the assumption of univariate normality was checked with three tests (Kolmogorov–Smirnov, Cramer–von Mises, and Anderson–Darling) [40].

As the variables showed large departures from normal distribution, they were transformed to normal scores ($y_i$) using Blom's formula [41]:

$$y_i = \Phi^{-1} \frac{(r_i - 3/8)}{(n + 1/4)} \quad (1)$$

where $\Phi^{-1}$ is the inverse cumulative normal (PROBIT) function, $r_i$ is the rank of the $i^{th}$ observation, and $n$ is the number of observations that have non missing values for the ranking variable.

Once the assumption of normality for the transformed variables was verified, multivariate normality was assumed for the multitemporal and multi-band data set of UAV data.

3.  Testing the sensitivity of multi-band data to the infection severity level. Univariate (ANOVA) and multivariate (MANOVA) analyses of variance were carried out on the UAV data, the former to test the hypothesis that the class means for each quantitative variable (band reflectivity) were equal, whereas the latter to compare multivariate class means across several variables. Four multivariate statistical tests were used: Wilks' lambda, Pillai's trace, Hotelling–Lawley trace, and Roy's maximum root [42–47].

4.  Determination of the parametric discriminant model. Assuming that each severity class had a multivariate normal distribution, firstly, a parametric method was performed, aimed at developing a discriminant mathematical function or classification, which best separated between the two classes of severity [44]. The classification criterion can be a linear function, assuming the same variance–covariance matrix of responses across the severity classes, or quadratic, assuming each class with a unique variance structure. A chi-square test of equal variance was then performed [43]. Using Bayes theorem, the posterior probability of each observation belonging to each class was calculated, taking into account the prior probabilities of the classes [45,46]. Each observation was placed in the class to which it had the highest posterior probability to belong. Therefore, as the final product of the model, for each observation the most probable class was assigned together with its posterior probability, which can be considered as a measure of the uncertainty associated with such assignation or prediction.

5.  Determination of non-parametric discriminant model. Since multivariate normality might not be fully satisfied, a non-parametric approach was also estimated to be compared with the parametric one. Non-parametric discriminant methods are based on non-parametric estimates of class-specific probability densities. The non-parametric kernel method uses a fixed radius (r) and a specified kernel (k), which can be uniform, normal, Epanechnikov, biweight or triweight, to calculate the kernel density in each class [44]. The value of r and type of kernel, called smoothing parameters, determine the degree of irregularity in the estimate of the density function. Small values of r produce jagged density estimates, whereas large values produce smoother density estimates. Therefore, for each type of kernel, several (ten on average) r values were tested, and the optimal set of the smoothing parameters, which minimizes the error rates, was chosen.

6.  Comparison between the two models. The performance of each model was evaluated using cross-validation [45,46]. Cross-validation uses n-1 out of n observations to determine the discriminant function for the classification of the one observation left out. This is repeated for each of the n observations. The error-rate estimates were calculated by counting the number of misclassified observations; the class-specific error-count estimate was determined as the proportion of misclassified observations in the class. [47–49]. The overall error rate was calculated as a weighted average of the individual class-specific error-rate estimates by using the prior probabilities as the weights [50–54].

7. Graphical display of infection status. Canonical discriminant analysis was also performed to extract one (number of classes (2) minus 1) linear combination of the quantitative variables, called canonical variable, which best revealed the differences between the classes and had the highest possible multiple correlation with the classes. The standardized canonical coefficients were estimated, which express the partial contribution of each quantitative variable (band) to the canonical variable, and were then used to interpret its meaning.

8. Prediction phase. The better classification model was then used in the prediction phase, by determining the more likely severity class for an independent data set not used in the previous phase of construction of the model. In particular, eight plants in the Torchiarolo field, for which the visual surveys were missing, were used for severity class prediction on August 2019. The estimated posterior probability of the predicted class was also provided as a measure of prediction uncertainty.

All discriminant analyses were carried out with the DISCRIM procedure of SAS (SAS University edition, release 9.4).

An overview of the proposed combined geostatistical-discriminant approach, for jointly analyzing visual inspection data with UAV multitemporal and multi-band data, is shown in Figure 7. The flowchart illustrates the methodology defined for: extracting crown from UAV data, merging visual data with UAV data taking into account change of support, predicting the more probable severity class from UAV data, producing a synthetic map of plant status using the canonical variable and, finally, displaying the more likely infection severity class per sector of each plant with the associated prediction probability as a measure of its uncertainty.

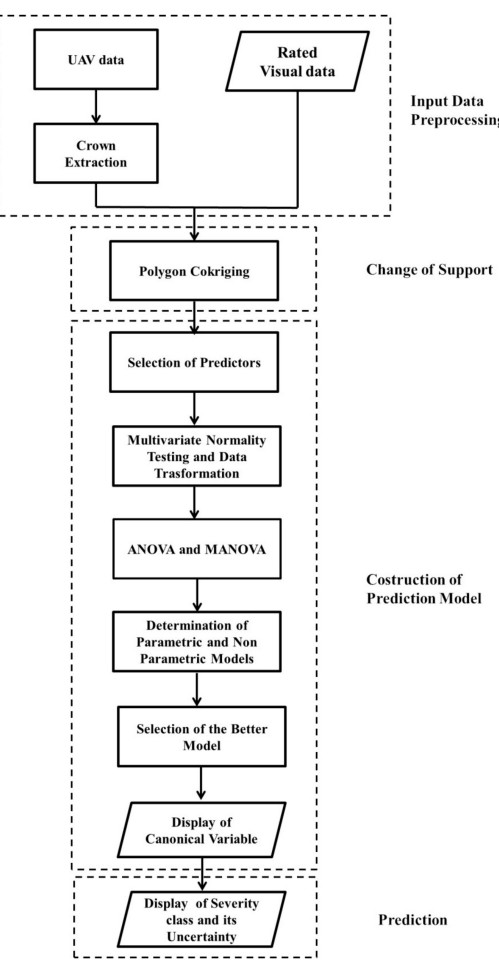

**Figure 7.** Flowchart of the proposed geostatistical-discriminant approach.

## 3. Results

### 3.1. Geostatistical Analysis

In order to apply the change of support for UAV data, an isotropic linear co-regiona lization model (LMC) was adapted to the multitemporal drone data set for each field, since no significant differences between the directional variograms were detected. In Table 1 the characteristics of the LMCs, fitted to the experimental variograms of the drone data for the three fields, are reported. In the supplementary material (see Supplementary Materials Figures S1–S3), there are the graphs of the experimental variograms with the fitted models, together with the partial sill matrices (covariance matrices) relative to each spatial structure (type of mathematical model).

**Table 1.** The characteristics of the linear models of co-regionalization (LMCs).

| Field | Type of Model | Range (m) | Spatial Variance Explained (%) |
|---|---|---|---|
| *Re* | Nugget effect | - | 32 |
| | Cardinal sinus | 18.64 | 68 |
| *Fella* | Nugget effect | - | 37 |
| | Cubic | 2.42 | 21 |
| | Cardinal sinus | 31.18 | 42 |
| *Torchiarolo* | Nugget effect | - | 54 |
| | Cardinal sinus | 22.23 | 37 |
| | Spherical | 88.62 | 9 |

Field Re—in the spatial dependence model, the structured component predominates over the spatially uncorrelated error (nugget effect), which remains, however, high due to high intra-plant spatial variability at centimeter scale, as monitored by UAV. The sinusoidal character of the structured component is due to the discontinuous nature of the variability (by units represented by the trees) with an influence scale corresponding to the average distance between the centroids of the trunks.

Field Fella—also for this field, the structured component is predominant on the one spatially uncorrelated. However, in this case, the former is further split into two sub-components: the one related to the short-range within-plant variation and the sinusoidal one due to the discontinuous nature of an olive grove. It should be noted that, for this particular plant arrangement, the influence scale of each tree is about double that of the previous field.

Field Torchiarolo—for this field, the not spatially structured component predominates over the structured one, which is split into two sub-components: the major one associated with the discontinuous structure of the olive grove and the minor one related to the intrinsic variation of the field at longer range. Figure 8 shows, for each field, the multitemporal map degraded to the quadrant support of the red-edge reflectivity, as an example. Similar maps were also provided for the other three bands together with their deviation standards (not shown). The red-edge was chosen because, as already discussed before, it proved to be a valid indicator of the physiological status of the plant, as any stress generally induces a lowering of reflectivity in NIR and a shift towards blue. Therefore, plants with a low reflectivity in the red-edge might be symptomatic of *Xfp* infection.

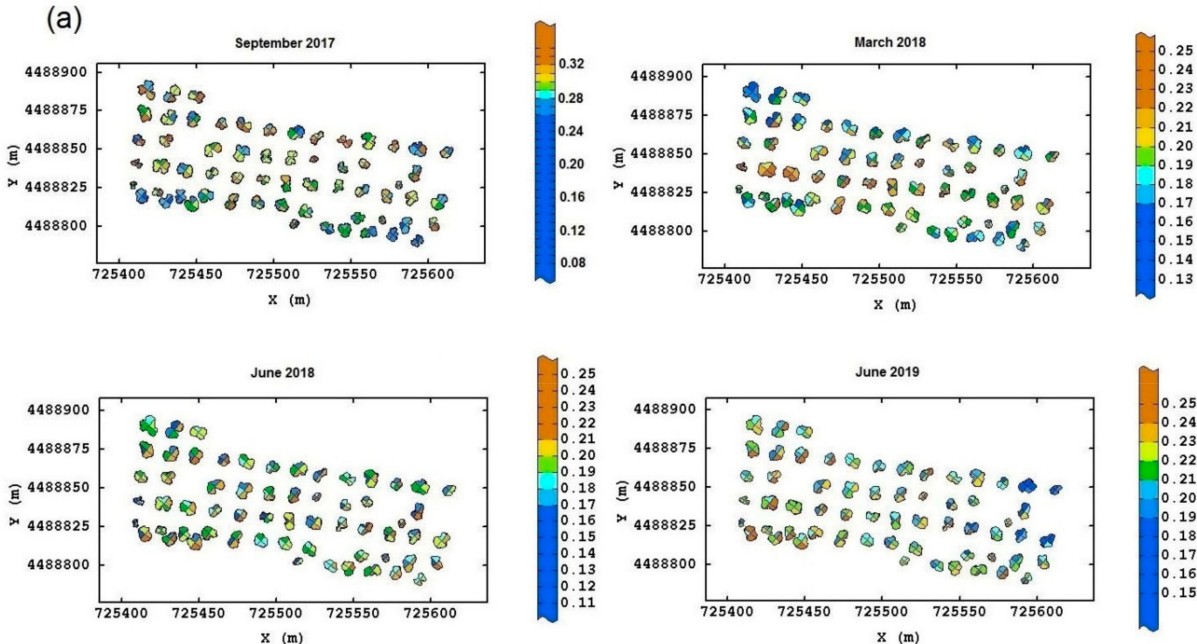

Coordinate System: WGS 1984 UTM Zone 33N
Projection: Transverse Mercator
Datum: WGS 1984

**Figure 8.** *Cont.*

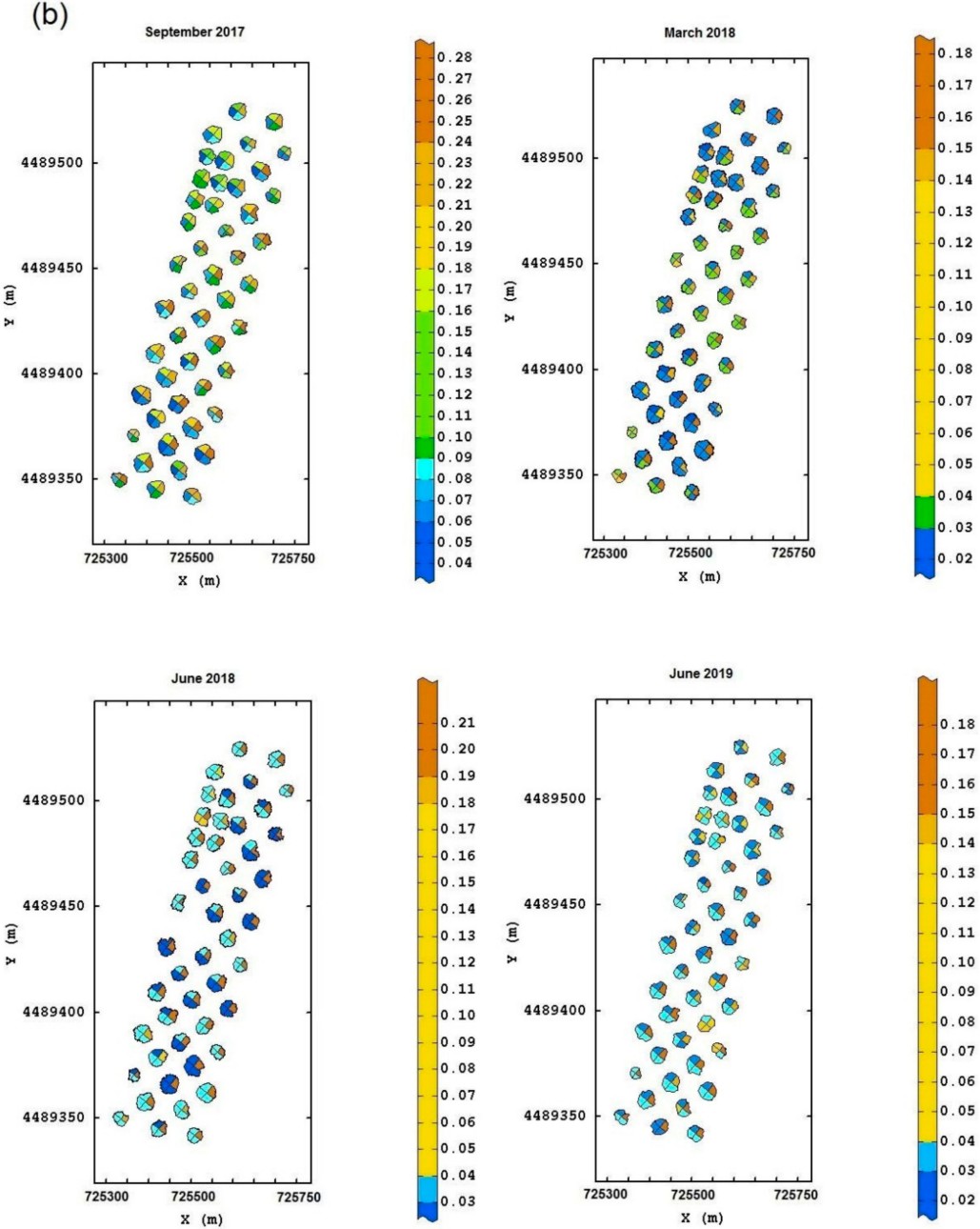

Coordinate System: WGS 1984 UTM Zone 33N
Projection: Transverse Mercator
Datum: WGS 1984

**Figure 8.** *Cont*.

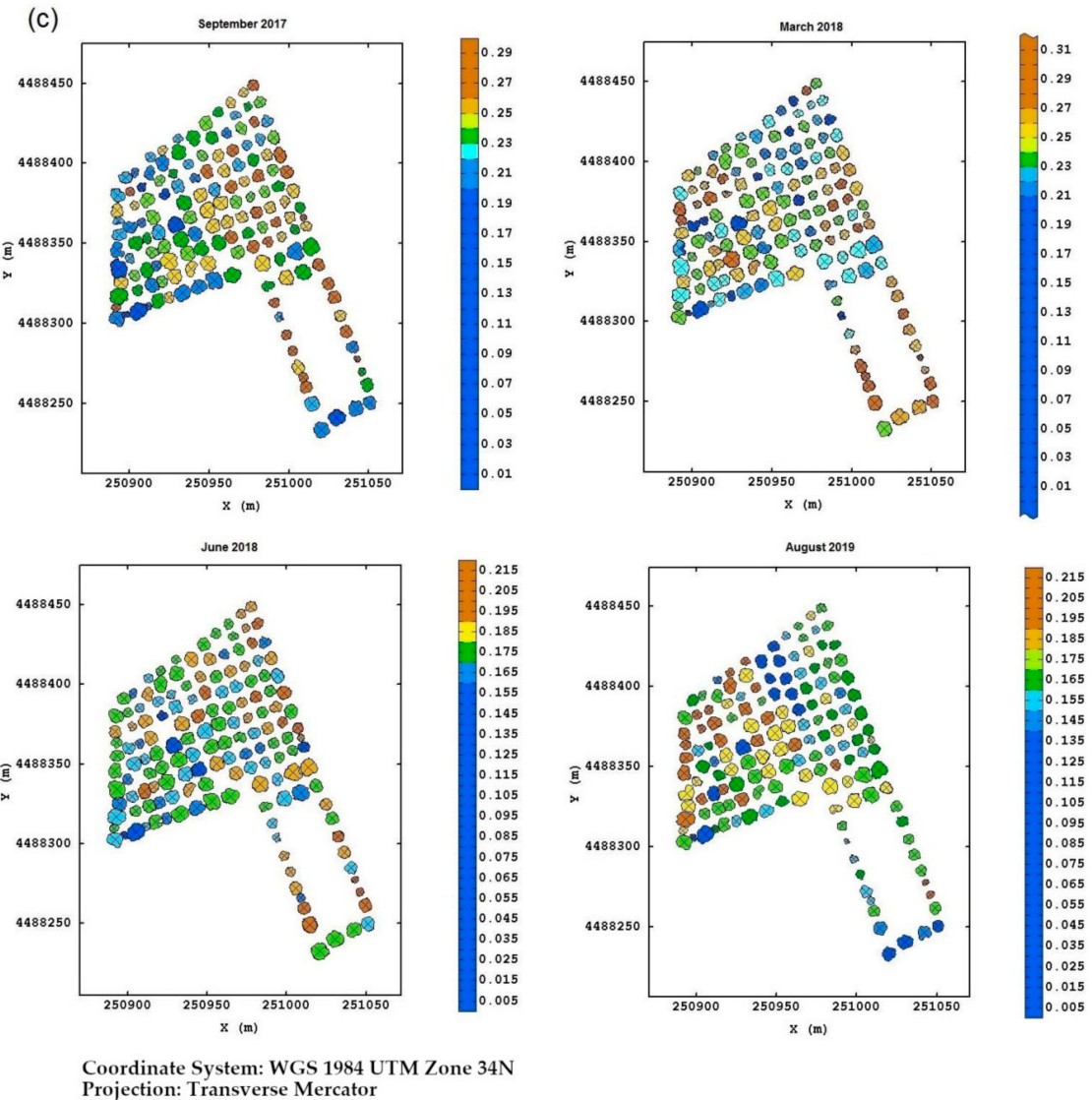

**Figure 8.** (**a**) Temporal red-edge reflectivity maps at plant quadrant level for the Re field. In this figure, and the following ones, the color scale is on isofrequency classes to enhance the differences. (**b**) Temporal red-edge reflectivity maps at plant quadrant level for the Fella field. (**c**) Temporal red-edge reflectivity maps at plant quadrant level for the Torchiarolo field.

Wanting to highlight a temporal trend in the evolution of the disease, in the Re field, the most stressed plants were first those on the south edge, and then those on the north edge and northwest corner at the first two monitoring times. At the third date, there was a sort of recovery of the plants, while in June 2019, it was apparent that the plants of the eastern section showed evident symptoms of disease.

As for the Fella field, the evolution of the disease was clearer. While on the date of the first flight the plants in the south, and partly in the north, were showing clear signs of suffering, by March 2018 almost all of the plants showed wilting symptoms.

For the Torchiarolo field, the situation appeared to be even more complex due to the high variability between the plants within the field. While on the first three monitoring dates, the plants on the west side were the most extensively and severely injured, in August 2019, some of these plants seemed to have recovered considerably. This particular behavior should be attributed to particular manifesting of the symptoms of the disease and to the intense production of new shoots as a plant defense mechanism, so that the trees might appear less affected, and seem to recover from the disease. Moreover, although wilted

leaves remain attached to the twigs, they tend to fall to the ground after a heavy rain or strong wind, thus masking the real disease severity on the plant.

### 3.2. Statistical Analysis

The study data set included 4095 observations, of which 2373 in class 0, with a proportion (prior probability) equal to 0.58, and 1722 in class 1, with a proportion equal to 0.42. Although the data set was slightly unbalanced towards class 0, it was considered an appropriate sampling since the analysis focused on the detection of infection in the very early stages.

As from the application of the procedure STEPDISC to the UAV data set all the eight variables resulted significant at 0.10 probability level, they were included in the successive statistical procedures.

None of the eight variables satisfied the univariate normal assumption based on Kolmogorov–Smirnov, Cramer–von Mises, and Anderson–Darling ($p < 0.01$) (Table 2). Therefore, all of the variables were transformed into ranks by using the Blom transformation, which was very effective in transforming the raw skewed distributions of all variables into normal ones. Multivariate normality within each class was then assumed and from this moment on, all analyses are referred to the ranks.

**Table 2.** Normality tests for the radiometric variables.

| Variable | D* | | W-Qu* | | A-Qu* | |
|---|---|---|---|---|---|---|
| | *Statistics* | *Probability* | *Statistic* | *Probability* | *Statistic* | *Probability* |
| *GREEN* | 0.281019 | <0.0100 | 105.9703 | <0.0050 | 585.016 | <0.0050 |
| *RED* | 0.21957 | <0.0100 | 56.25436 | <0.0050 | 331.2315 | <0.0050 |
| *RED-EDGE* | 0.164006 | <0.0100 | 17.94999 | <0.0050 | 123.1129 | <0.0050 |
| *NIR* | 0.066529 | <0.0100 | 2.045712 | <0.0050 | 10.74146 | <0.0050 |
| *GREEN_std* | 0.442217 | <0.0100 | 157.4172 | <0.0050 | 830.784 | <0.0050 |
| *RED_std* | 0.391301 | <0.0100 | 126.0456 | <0.0050 | 728.0533 | <0.0050 |
| *RED-EDGE_std* | 0.370405 | <0.0100 | 111.0805 | <0.0050 | 552.6192 | <0.0050 |
| *NIR_std* | 0.281603 | <0.0100 | 65.14581 | <0.0050 | 323.1493 | <0.0050 |

D* = Kolmogorov–Smirnov, W-Qu* = Cramer–von Mises, A-Qu* = Anderson–Darling.

The preliminary analysis of ANOVA showed that the means of the two severity classes were significantly different for all the variables at the probability level of $p < 0.0001$, whereas they were not differentiated only by the standard deviation of the rank of red reflectivity (Table 3).

**Table 3.** Results of ANOVA on rank variables. R before the name of the variable means rank.

| Variable | F Value | Probability |
|---|---|---|
| R_GREEN | 134.09 | <0.0001 |
| R_RED | 24.07 | <0.0001 |
| R_RED-EDGE | 174.09 | <0.0001 |
| R_NIR | 106.90 | <0.0001 |
| R_GREEN_std | 50.57 | <0.0001 |
| R_RED_std | 0.84 | 0.3580 |
| R_RED-EDGE_std | 90.59 | <0.0001 |
| R_NIR_std | 16.86 | <0.0001 |

The MANOVA analysis consistently showed high statistical significance with the four tests (Table 4). These results indicate that the two severity classes exhibited different spectral behavior not only at each individual band but also overall.

**Table 4.** Results of MANOVA on rank variables.

| Test | Statistic Value | F Value | Probability |
|---|---|---|---|
| Wilks's lambda | 0.931 | 37.74 | <0.0001 |
| Pillai's trace | 0.069 | 37.74 | <0.0001 |
| Hotelling–Lawley's trace | 0.074 | 37.74 | <0.0001 |
| Roy's maximum root | 0.074 | 37.74 | <0.0001 |

Having assumed multivariate normal distribution within each class, a quadratic discriminant analysis was performed since the Chi-Square value of the test of equal variance was statistically significant ($p < 0.1$).

The results of the cross-validation showed an overall accuracy of 0.64, while that of class 0 (producer's accuracy) was 0.59, and the one of class 1 was 0.71 (Table 5). Actually the accuracy was quite low for class 0, which was the focus of the work. This is also confirmed by the error rates or misclassification probabilities that are rather high, especially for class 0 (Table 6).

**Table 5.** Confusion Matrix for *X. fastidiosa* subsp. *pauca* severity classes using the quadratic discriminant classification with the absolute counts and the accuracies.

| | | Ground Truth | | Total # of classified samples | User's accuracy |
|---|---|---|---|---|---|
| | | 0 | 1 | | |
| | 0 | 1405 | 498 | 1903 | 0.74 |
| | 1 | 968 | 1224 | 2192 | 0.56 |
| **Classification Results** | Total #of ground truth samples | 2373 | 1722 | | |
| | Producer's accuracy | 0.59 | 0.71 | | 0.64 |

**Table 6.** Error rates for *X. fastidiosa* subsp. *pauca* severity classes using the quadratic discriminant classification.

| Class | 0 | 1 | Average |
|---|---|---|---|
| **Error Rate** | 0.41 | 0.29 | 0.36 |

As a cause of error might have been the assumption of multivariate normality, which might not be actually fully satisfied, a non-parametric approach was alternately performed and after several trials on the smoothing parameters the optimal choice was: normal kernel with r equal to 0.3.

Tables 7 and 8 show the results of cross-validation for the non-parametric method. Even if the overall results did not change appreciably (overall accuracy = 0.69 and mean error rate = 0.31), nevertheless the situation is completely reversed for the two classes. Fairly good results were obtained for the early detection class with a class accuracy (producer's accuracy) of 0.77 and a misclassification probability of 0.23.

**Table 7.** Confusion Matrix for *X. fastidiosa* subsp. *pauca* severity classes using the non-parametric method with the absolute counts and the accuracies.

| | | Ground Truth | | Total # of classified samples | User's accuracy |
|---|---|---|---|---|---|
| | | 0 | 1 | | |
| Classification Results | 0 | 1835 | 730 | 2565 | 0.72 |
| | 1 | 538 | 992 | 1530 | 0.65 |
| | Total #of ground truth samples | 2373 | 1722 | | |
| | Producer's accuracy | 0.77 | 0.58 | | 0.69 |

**Table 8.** Error rates for *X. fastidiosa* subsp. *pauca* severity classes using the non-parametric method.

| Class | 0 | 1 | Average |
|---|---|---|---|
| Error rate | 0.23 | 0.42 | 0.31 |

Finally, a canonical discriminant analysis was carried out in order to produce a dimension reduction of the eight quantitative variables to a single canonical variable and to highlight those variables that better discriminated between the two classes. The correlation coefficient, although significant ($p < 0.01$), was rather low, equal to 0.26. However, the two classes were well separated from the canonical variable, reporting mean values equal to 0.23 and $-0.32$, for class 0 and 1, respectively. Table 9 shows the canonical coefficients standardized within the classes, being the variances not homogeneous, from which it can be deduced that the most positively influential variables were red-edge and red, the former indicative of the overall physiological status of the plant, while the latter more closely related to the chlorophyll function. On the reverse side, the red standard deviation weighted negatively, which can be interpreted as a larger variability in the most stressed plants due to the concomitant presence of foliage with different degree of desiccation.

**Table 9.** Within-class standardized coefficients of the canonical variable.

| Variable | Coefficients |
|---|---|
| R_GREEN | 0.034 |
| R_RED | 0.652 |
| R_RED-EDGE | 0.885 |
| R_NIR | −0.247 |
| R_GREEN_std | 0.171 |
| R_RED_std | −0.689 |
| R_RED-EDGE_std | 0.364 |
| R_NIR_std | −0.154 |

In Figure 9 the map of the canonical variable, corresponding to the monitoring of the Torchiarolo field in March 2018, is shown as an example, where it is evident that the plants in the south–west corner were suffering for some kind of stress.

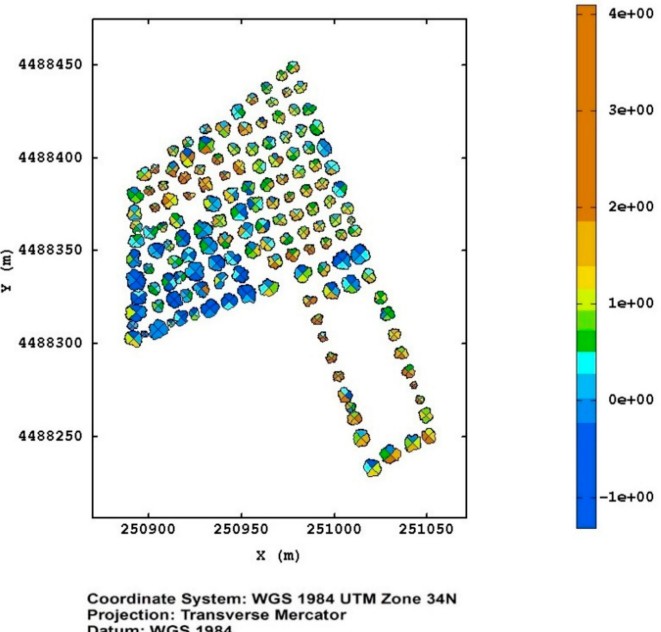

**Figure 9.** The map of the canonical variable for the Torchiarolo field on March 2018.

In Figure 10, it is reported, as an example, an output of the prediction model concerning eight plants in the Torchiarolo field that had not been visually rated by the operator at the time of the flight of August 2019. In this case, we are concerned with plants in an advanced status of drying; however, we can note that not all of the predictions have the same degree of uncertainty. While predictions with a probability of 0.98 can be considered reliable, those with probabilities of 0.54 are extremely uncertain.

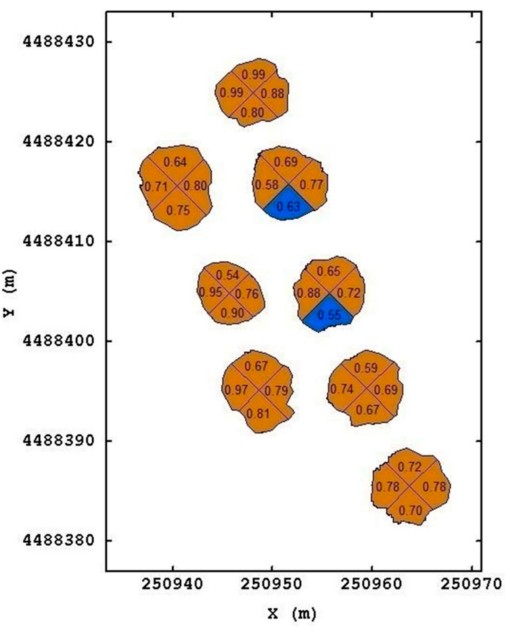

**Figure 10.** The prediction of the more probable severity class, together with its posterior probability at plant quadrant level, for eight plants at Torchiarolo field on August 2019. The brown color represents class1, blue color represents class 0.

## 4. Discussion

The geostatistical analysis on UAV data has evidenced the extreme variability that characterizes the olive trees, even within individual plants. This then makes it hard to extract clear evidence of the occurrence of the disease. Therefore, only a multivariate and multitemporal statistical analysis could hopefully produce more reliable predictions.

Several reasons may have caused the poor results of discriminant analysis with the parametric method: a sub-optimal choice of the radiometer wavelengths, more confusing factors that may have contributed to the drying of the vegetation, but not directly ascribable to *Xfp* infection, and a different appreciation of the canopy quadrant support from ground at man height rather than from above the plants at the height of UAV flight.

Another cause of error might have been the assumption of multivariate normality that was not actually fully satisfied. Therefore, a non-parametric approach was alternately performed. The rather low value of accuracy confirms the high stochasticity, characterized by short-range variability, as already observed in the geostatistical analysis of spatial dependence (fitting of LMCs).

The undoubtedly encouraging results obtained with the non-parametric method can be further improved with an appropriate choice of the classification method, an optimal calibration of its parameters, and a preference for the hyperspectral (rather than multispectral) sensor. The former indeed allows the choice of quite narrow-band wavelengths that are particularly discriminating *Xfp*-infected plants, as it was previously verified by other authors [16].

The canonical variable can be used as an efficient and synthetic, although relative, spatial indicator of the overall health of the olive tree. According to its structure (Table 1), high and positive values denote luxuriant and homogeneous enough foliage with good chlorophyll function, while lower and negative values indicate clear symptoms of stress and drying of the foliage.

A spatial map of this variable (Figure 9) could be used as an efficient tool for decision-making, and support to address the monitoring activity, and direct the scouting for ascertaining the effective presence of the bacterium in the pre-selected plants by means of the molecular diagnostic techniques. From the examination of Figure 9, it can be noted that, leaving aside the plants evidently stressed in the south–west corner, some plants in the north–west and south–east corners of the upper portion of the field showed, especially in some sectors, incipient stress conditions. These plants should be given special attention with the perspective of avoiding the spread of the infection.

Another way of using UAV data as a decision making tool is to provide the prediction of the most probable severity class together with its posterior probability, which is a measure of the degree of uncertainty, and then of the level of risk associated with a decision based on that prediction.

Therefore, priority should be given to those plants for which prediction is more uncertain.

In addition, scouting should be intensified precisely where the prediction is most uncertain.

## 5. Conclusions

In this study, a combined procedure, based on an integrated use of some geostatistical and discriminant analysis techniques, was developed for the semi-automatic classification of Xylella-infected olive plants showing different symptoms severity. High spatial resolution imagery, recorded with a multispectral camera installed on board of an unmanned platform, was acquired over three olive orchards differing in plant characteristics, age, and planting system on three different dates. The radiometric data were firstly degraded to the support of visual assessment, using geostatistical techniques, after that a quadratic discriminant analysis and a non-parametric method were applied to discriminate between two severity classes, with a focus on early detection. The non-parametric approach classified better the trees at initial and low severity levels compared with within-class accuracy of 77%.

The proposed approach, although including several geostatistical and statistical techniques implemented using commercial software, uses standard procedures that could be easily rewritten with free software, such as R [55], geoR [56], and Qgis [57]. An exception is polygon cokriging, which, to our knowledge, is only implemented in the commercial ISATIS software. This procedure is actually a modification of block cokriging in the calculation of the areal spatial covariance function, due to the irregular and variable shape of the crown polygons (blocks). However, its implementation using a common programming language should not present particular difficulties.

Therefore, we believe that the realization of a decision support system (DSS), aimed at identifying plants that might be asymptomatic or in the early stages of Xylella infection, could be easily achieved after a suitable informatization of the approach illustrated. However, it is worth underlining, the prediction of the most probable level of infection severity cannot be done in real time, but after the UAV data acquisition on the area of interest.

The system is extendable to any type of data recorded, not only in remote (UAV, airplane, satellite), but also in proximal sensing. It is also flexible enough to include other classification techniques, such as those of machine learning.

The results are promising and seem to encourage the application of UAV technology in the early detection of Xylella infection, and we expect that UAV spectral sensing systems will become a powerful tool in disease detection, not only for researchers, but also for service providers and even farmers.

The results of numerous applications of proximal, UAV-borne, air-borne, and satellite-borne imaging sensors seem to indicate their complementary rather than competitive use, even over the same area. This is because many factors (costs, spatial resolution, temporal frequency, areal coverage, type of features to be extracted, wavelength range, etc.) may influence this choice [14].

For this reason, some authors propose to combine the images acquired at the various spatial scales with different sensors, using a data fusion approach, in order to improve the ability of stress detecting [48].

Therefore, disease monitoring with UAV data is of limited value when used individually. On the contrary, it acquires greater value if used in a wider knowledge system that integrates information of different (physiological, meteorological, and agronomic) type, as well as data from other sensors through geostatistical techniques of data fusion [27]. To investigate the full potential of UAVs in disease detection, some important future research priorities still remain, such as defining the scale at which to estimate the severity of the infection: leaves, plants, quadrats or quadrants?

Further (different) methods of automatic disease assessment should be compared, including discriminant analysis and various techniques of machine learning.

Finally, it is urgent to underline the importance of defining an appropriate sampling strategy in order to efficiently identify plants in the early stages of infection, thus allowing the operator to intervene in a timely manner. In this, the UAV data can give a valid support since, as it has been extensively demonstrated in this work, they can aid in classifying plants with different levels of symptom severity.

**Supplementary Materials:** The following are available online at https://www.mdpi.com/2072-4292/13/1/14/s1, Figure S1: LMC for UAV data of the Re field, Figure S2: LMC for UAV data of the Fella field, Figure S3: LMC for UAV data of the Torchiarolo field.

**Author Contributions:** Conceptualization, A.C., F.N., A.B., G.G.; methodology, A.C., A.B., F.N.; investigation, A.C., A.B., S.S.; data curation, E.B., R.Q., F.Q., G.G., F.N., M.R.M., N.A.R., C.R., S.R., S.S., I.A., V.S.; writing—Original draft preparation, A.C., A.B., S.S., F.N.; writing—Review and editing, A.C., A.B., S.S., F.N.; visualization, A.B., S.S. All authors have read and agreed to the published version of the manuscript.

**Funding:** This research was funded by Apulia Region.

**Acknowledgments:** Project "XylMap—Identification of CoDiRO diffusion dynamics after analysis of progression mechanisms and development of enhanced monitoring and mapping tools and methods" was financed by the Apulia Region (Italy) with reference to DD n. 494 of 14/10/2015 and n. 278 of 9/8/2016 (Cod. A).

**Conflicts of Interest:** The authors declare no conflict of interest. The funders had no role in the design of the study; in the collection, analyses, or interpretation of data; in the writing of the manuscript, or in the decision to publish the results.

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
