# Peer review of "Semi-Automatic Method for Early Detection of Xylella fastidiosa in Olive Trees Using UAV Multispectral Imagery and Geostatistical-Discriminant Analysis"

_remotesensing, doi:10.3390/rs13010014_

Round 1
Reviewer 1 Report
This paper employed multispectral images (4 bands: R, G, Red-edge, NIR) captured by UAV to classify Xylella disease infection into 2 levels in olive trees, and implemented experiments at three locations in Italy.
In general, the paper is a bit rough in both technical descriptions and scientific English writing.
In the aspect of technical presentation:
- The authors first categorized disease severity into six levels and then stated that they only aimed to solve a binary classification problem, which was explained as a reason for early disease detection. Such task transformation reason is a bit weak and the experimental results for such relatively simply binary classification (accuracy: 0.77) were a bit low.
- The author used traditional methods based on some geostatistical and discrimination analysis techniques. I would like to see the performance and comparative experiments for this task by deep convolution neural network (CNN).
- The authors stated that they focused on disease detection at an early stage. However, I cannot find the descriptions about the disease development and the definition of ‘early detection’ in this case.
- The methods used by the authors involves some hyper-parameters, which were not described in detail thus may affect experimental reproductivity.
In the aspect of scientific presentation:
- Figure 8, Table 5: The figures and tables’ are not consistent with commonly-used formats. There is usually one main caption and written as Figure No. or Table No. following by its description, rather than writing Figure/Table each time for each subfigure/subtable.
- There are many punctuation mistakes in the paper, e.g., L160: no punctuation before two independent sentences, L201: should be a full stop after a complete sentence, etc.
- Figure 3a and Figure 4b: 1) why two subfigures are named with two figure number? 2) please remove the software interface part for the figures.
- The sentence “Error! Reference source not found.)” appeared several times in the paper. What do they mean??
Author Response
Dear Reviewer,
We realise that there was an error in the phase of acquisition of the manuscript by the publisher and that this had a negative impact on the readability and interpretation of the text.
The revised manuscript (pdf and word), and responses to comments, are attached.
Regards
Antonella Belmonte
Technical presentation
Point 1: The authors first categorized disease severity into six levels and then stated that they only aimed to solve a binary classification problem, which was explained as a reason for early disease detection. Such task transformation reason is a bit weak and the experimental results for such relatively simply binary classification (accuracy: 0.77) were a bit low.
Response 1: As we have explained in the text it is necessary to act in the very early stages of the infection, as any delay would preclude the attempt to slow down the progression of the epidemic and increase the infection risk for the surrounding plants. This aspect is particularly relevant for a pathogen such as Xylella fastidiosa whose latency period on olive trees (i.e. the time lapse from the infection to the symptoms appearance) can last 12-18 months. Moreover, from a statistical point of view, the 6 classes did not differ statistically from the null hypothesis of being attributable to chance.
Point 2: The author used traditional methods based on some geostatistical and discrimination analysis techniques. I would like to see the performance and comparative experiments for this task by deep convolution neural network (CNN).
Response 2: In the introduction and in materials and methods we have justified our methodological choices due to: 1) determination of a mathematical model; 2) presence of the spatial correlation between the observations; 3) taking into account the change of support for the variables used to construct the model. In our opinion, the CNN method does not consider these issues. Moreover, it was not our objective to compare the proposed approach with others. CNN is advantageous in other experimental contexts (big data, automation, etc.).
Point 3: The authors stated that they focused on disease detection at an early stage. However, I cannot find the descriptions about the disease development and the definition of ‘early detection’ in this case.
Response 3: As recalled under point 1, the latency period (i.e. the period without any symptoms appearance after the bacterial infection) can last 12-18 months. During this period, the concentration of the bacterium into the olive xylem increase, as well as the infectious potential of the plant. Therefore, the “ideal” early detection moment would be located into the latency period, when the symptoms are still not visible. However, this is not possible, being the monitoring activity issued by the regional Phytosanitary Authority based on the “visual inspection” of the plant, i.e. searching for wilting symptoms on the canopy of the tree. This is the reason why we have made it clear in the text that the response variable is based on visual estimation of the degree of desiccated canopy. We have clearly defined that for us early detection corresponds to a proportion of canopy desiccation of 0.05 for each sector. This proportion has a heuristic value and is determined on the basis of the consolidated experience of expert evaluators, considering that lower values can escape the resolving power of the human eyes.
Point 4: The methods used by the authors involves some hyper-parameters, which were not described in detail thus may affect experimental reproductivity.
Response 4: We do not understand to which hyper-parameter the reviewer is referring. Please be more precise.
Scientific presentation
Point 1: Figure 8, Table 5: The figures and tables’ are not consistent with commonly-used formats. There is usually one main caption and written as Figure No. or Table No. following by its description, rather than writing Figure/Table each time for each subfigure/subtable.
Response 1: We have explained at the beginning that there was a mismatch between the submitted version and the one acquired by the publisher with regard to the figures. In the modified version we used a single caption for multiple figures and different caption for tables.
Point 2: There are many punctuation mistakes in the paper, e.g., L160: no punctuation before two independent sentences, L201: should be a full stop after a complete sentence, etc.
Response 2: We checked the punctuation.
Point 3: Figure 3a and Figure 4b: 1) why two subfigures are named with two figure number? 2) please remove the software interface part for the figures.
Response 3: The figures do not correspond to the original version. We have put the correct ones. We have removed the software interface part for the figures.
Point 4: The sentence “Error! Reference source not found.)” appeared several times in the paper. What do they mean??
Response 4: We have removed the sentence: “Error! Reference source not found.” The reason was explained above.
Reviewer 2 Report
The manuscript has improved greatly, although some aspects remain to be polished. I recommend a minor revision, although the volume of changes to be made should be considered and accomplished accordingly. I list them below.
The reviewers should not act as proofreaders, I demand an in-depth and strong review in this regard. Examples:
- Inconsistent spelling: both North American and British spellings are used in the manuscript. Both are acceptable, but it’s best to be consistent.
- Error! Reference source not found. Also found 4 times more!
- Xfp acronym is used emphasised (italic) and not. Please, be consistent.
- There are two figures 1.
Minor comments:
- Rededge? I would say either red-edge or red edge.
- Multi-spectral or multispectral? The latter the better, but please, be consistent.
- Figure 1 (the first one). Please add the projection information. It is happening the same from Figure 8 onwards.
- Figure 1 (the first one). Please, consider redrawing the scale, it is not visible.
- Figure 3. Please link both views (a and b), it would be easier for the reader to understand.
- Figure 4. I was wondering if figure 4 was copied and pasted from another source. It is weird to find it labelled as “Figure 6” on the top.
- There are two figures 4.
- There are two figures 5.
- Figure 5. Please, improve the quality of the diagram, it seems to be a snapshot.
- All figures should be referenced accordingly. I have found many mistakes in this regard.
It is a shame that the quality of this work is diminished by a very poor presentation that needs to be worked on. The content has improved a lot, and so has the discussion. However, this should be reviewed by a professional proofreader to prevent this work from going back on reviewers. It becomes very difficult to read in the current state.
Author Response
Dear Reviewer,
We realise that there was an error in the phase of acquisition of the manuscript by the publisher and that this had a negative impact on the readability and interpretation of the text.
The revised manuscript (pdf ) is attached.
Best Regards
Antonella Belmonte
Point 1: 
 Inconsistent spelling: both North American and British spellings are used in the manuscript. Both are acceptable, but it’s best to be consistent.
Response 1: We did a British spelling check and some errors were found, but only using a tool on line. For Word, the text is correct.
Point 2: Error! Reference source not found. Also found 4 times more!
Response 2: We have removed the sentence. However, the reason is in the mismatch between the manuscript submitted and the one acquired from the publisher.
Point 3: Xfp acronym is used emphasised (italic) and not. Please, be consistent.
Response 3: We have modified it.
Point 4: There are two figures 1.
Response 4: The reason was explained above.
Minor comments:
Point 1: Rededge? I would say either red-edge or red edge.
Response 1: We have modified it, using red-edge.
Point 2: Multi-spectral or multispectral? The latter the better, but please, be consistent.
Response 2: We have modified it, using the latter.
Point 3: Figure 1 (the first one). Please add the projection information. It is happening the same from Figure 8 onwards.
Response 3: The projection information is added in the figures.
Point 4: Figure 1 (the first one). Please, consider redrawing the scale, it is not visible.
Response 4: We have modified the figure 1 and redrawn the scale.
Point 5: Figure 3. Please link both views (a and b), it would be easier for the reader to understand.
Response 5: We have linked both views (a and b) in one Figure 4 (the Figure 3 does not match the true version).
Point 6: Figure 4. I was wondering if figure 4 was copied and pasted from another source. It is weird to find it labelled as “Figure 6” on the top.
Response 6: We have improved the resolution of the figure, and its corrected numbering now is Figure 7. Therefore, the reference to figure 6 in the text is not correct, we have removed this reference. The explanation is again in the mismatch between the two types of figures.
Point 7: There are two figures 4
Response 7: Again, the explanation is in the noted mismatch.
Point 8: There are two figures 5
Response 8: See above.
Point 9: Figure 5. Please, improve the quality of the diagram, it seems to be a snapshot.
Response 9: We have improved the quality of the diagram that now corresponds to the Figure 7.
Point 10: All figures should be referenced accordingly. I have found many mistakes in this regard.
Response 10: See above for explanation.
It is a shame that the quality of this work is diminished by a very poor presentation that needs to be worked on. The content has improved a lot, and so has the discussion. However, this should be reviewed by a professional proofreader to prevent this work from going back on reviewers. It becomes very difficult to read in the current state.
We understand your disappointment and greatly regret for the inconvenient. However, the mismatch between the two versions of the manuscript was out of our control. We have repeatedly complained to the publisher about the disagreement between the version actually submitted to the journal and the one that was acquired by the publisher and sent to the reviewers. We realize that this has greatly prejudiced the readability of the text and therefore the judgment of the reviewers on the quality of our work.

This manuscript is a resubmission of an earlier submission. The following is a list of the peer review reports and author responses from that submission.
Round 1
Reviewer 1 Report
This study is to detect Xylella infection for olive trees at early stage based on UAV-carried 4-band(R,G,B,IR)multispectral images. Three olive groves that locate in south-eastern Italy during 2017 to 2019 were used in the experiment. Class accuracy of 0.77 were reported for the early detection task.
Technically speaking, the paper is lack of novelty. The proposed method is composed of different existed techniques and softwares, which will affect its robustness and adaptivity to other regions or disease varieties. Moreover, the method descriptions are somewhat unclear and detailed. Please describe each method part with more detailed description and intuitive illustration.
The authors stated that the images were captured at a height of 70m with a very high resolution of 6.6cm/pixel. However, the high resolution cannot be reflected in the illustrated images of Figure 2.
In L141, please explain the relationship between the degree of desiccation and two-class classification. It is somewhat sudden and confused for me when the authors introduced their ‘early’ and ‘not early’ classification standard by the degree of the desiccation without given any prior knowledge and information.
Reviewer 2 Report
- It's hard to see Figure 3. Correction is required.
- Is it possible to distinguish whether the effect of the red edge area is due to infection or lack of water and other factors?
Reviewer 3 Report
The paper entitled " Semi-automatic method for early detection of Xylella fastidiosa in olive trees using UAV multispectral imagery and geostatistical-discriminant analysis" presents a comprehensive work that has a strong novelty in terms of the possibilities of application of low-cost systems for the purpose of monitoring a disease that affects a growing number of regions in the world. Its applicability is of interest, but it isn't very easy to follow the steps carried out to make the process reproducible. This study is very well prepared and potentially well-received work by the audience, although it needs further major revision before being published.
Major comments:
English grammar should be revised. North American and British terms are used indistinctively (e.g. recognize, analyse, localization, centimetre, ...). Also in the usage of the/a, somehow there are some missing determiners where an article is needed.
It is challenging to understand the initial classification of 6 classes, and then combine them into 2. I imagine that it will be an attempt at simplicity, but it is difficult to understand whether it was a matter of looking for the set of groupings that behaved the best, or whether it was simply the result of chance or following some previously analysed protocol that has not been described. I would like to understand a little more why a particular classification has been considered and not another one.
The methodology concerning the extraction of the crown (segmentation) should be explained in more detail. Also, everything related to the subdivision of quadrants and radiometric measurements is complicated to follow with the current explanation.
The work revolves around an improved "Polygon cokriging" technique for the detection of Xylella; however, the explanation of the methodology is feeble, and some figures are missing to help understand the method better.
There are many statistical methods described in the results section that are not only not described in materials and methods but not even mentioned. The manuscript needs a substantial revision in this sense, it is very complicated to keep up with the flow, and in the results section, you can find previously undescribed (and unreferenced) methodologies, which makes it very difficult to understand. Likewise, regarding the spectral analysis on how the multispectral and not an ordinary camera is needed.
Discussion. It would be interesting to go deeper into the advantages and limitations of the use of drone with respect to airborne and satellite, so that the possibility of a real application of this methodology is really understood.
Minor comments:
Abstract.
Is the classification working at plot/pixel level? Does it classify trees or plots? In which way? It is unclear whether you rank damage level (severity) or affection (incidence).
L31. I would say "the Apulia region" instead
L35. "encourages" does not correspond to the plural subject.
Introduction
L43. It may be unclear what "This" refers to. Please consider rewriting the sentence to remove the unclear reference.
L48. "Consist of"
L80. "a the blue"?
Materials and Methods
L108. Could you please provide the exact location?
Fig.1. A scale or coordinates in each zoom level would be useful.
L137-139. Could you please describe the method of visual classification, or including a supporting reference? Also how you fixed the threshold
L140 “only two classes were considered”. Could you please explain the transformation between 6 levels to 2. Also, a figure which includes tree samples would be useful and maybe self-explained.
L148 What is the endurance of this drone with the payload?
L157 Which Pix4D version?
L167 Could you please explain more (apart from the references) about the impact of the “morphological filter” in this analysis?
L168 Which ENVI version?
L171 Which GIS were you using?
L179 Could you please specify the target degradation?
L197 Which SAS version?
Results
L255 Geostatistical Analysis. It is tough to follow the outcome. A table summarising the results would be needed.
Figure 3. I would suggest increasing the font size and figures distribution. It is hard to read/follow.
Table 1. is “std” standard deviation? With a negative value?
Conclusions
L417-419 Did you find any limitation due to the flight plan?
L421 You were using 2 classes for the classification, weren’t you? How can you describe it as classifying the severity? It sounds odd to me.